# Development and Characterization of Symbiotic Buffalo Petit Suisse Cheese Utilizing Whey Retention and Inulin Incorporation

**DOI:** 10.3390/foods12234343

**Published:** 2023-12-01

**Authors:** Rebeca Morais, Pedro Ivo Soares, Sinthya Kelly Morais, Suelma Oriente, Amanda Nascimento, Mylena Olga Melo, Francisca Moises Sousa, Mario Cavalcanti-Mata, Hugo M. Lisboa, Rennan Pereira Gusmão, Thaisa Abrantes

**Affiliations:** Food Engineering Department, Universidade Federal Campina Grande, Av. Aprígio Veloso 882, Campina Grande 58429-900, Paraíba, Brazilthaisa.abrantes@professor.ufcg.edu.br (T.A.)

**Keywords:** dairy products, prebiotic, probiotic, lactic acid bacteria, functional ingredients

## Abstract

This study presents the development and characterization of a novel buffalo Petit Suisse cheese, enhanced with symbiotic properties through an innovative whey retention method and incorporating inulin and xanthan gum. The research focused on assessing the cheese’s physicochemical properties, shelf life, lactic acid bacteria viability, syneresis behavior, and the impact of varying concentrations of functional ingredients. The addition of inulin and xanthan gum, following a design of experiments, significantly influenced the cheese’s texture and consistency. Higher inulin concentrations were associated with increased fermentation activity, as indicated by total titratable acidity, which showed an increase from 1.22% to 1.50% over a 28-day period, and pH levels that decreased from 3.33 to 2.96. The syneresis index varied across trials, with the highest reduction observed in trials with increased xanthan gum concentrations, effectively reducing syneresis to 0%. Lactic acid bacteria viability also showed notable variations, with the highest cell survival percentage reaching 107.89% in formulations with higher inulin and xanthan gum concentrations. These results underscore the importance of inulin and xanthan gum in enhancing the cheese’s microbial stability and textural quality. The study concludes that the strategic use of inulin and xanthan gum improves the nutritional profile of buffalo Petit Suisse cheese and optimizes its textural and sensory attributes.

## 1. Introduction

The contemporary food industry is witnessing a transformative era, characterized by a burgeoning consumer appetite for health-focused and environmentally sustainable products. This evolution is particularly impactful in the dairy sector, where there is a growing emphasis on nutrient-rich and ethically produced offerings [1]. Dairy companies globally are not only responding to this trend but also shaping it, introducing an array of innovative products that blend culinary delight with health benefits. These products, often enhanced with functional ingredients like rolled oats, chicory root extract, and inulin, not only enrich the dietary fiber content but also contribute to a broader spectrum of wellness advantages [2]

With its rich nutrient profile and unique physicochemical properties, Buffalo milk has emerged as a key player in this evolving dairy narrative [3]. Despite its numerous benefits, the processing of buffalo milk poses environmental challenges, primarily due to the generation of pollutant-heavy waste. Drawing inspiration from cleaner processing techniques employed in the bovine milk industry, such as the sustainable utilization of by-products like whey, offers a promising avenue for making buffalo milk processing eco-friendlier and more cost-effective [4].

Among various dairy products, Petit Suisse cheese, a creamy delicacy of French origin, is undergoing a transformative phase in its production process [5]. Traditionally, the production of petit Suisse involves coagulating milk with rennet and mesophilic bacteria, followed by centrifugation to separate the whey, which inadvertently leads to substantial product yield loss and waste generation. An innovative approach to circumvent these issues is the retention of whey in the cheese matrix, eliminating the need for draining and enhancing both yield and sustainability. This method also imparts a distinctive consistency to the cheese, setting it apart in texture and environmental impact [6].

Inulin, a soluble dietary fiber, has been increasingly used in dairy products, offering a range of benefits health benefits, including improved mineral absorption and immune function [7]. According to research by Karimi et al. (2015), inulin can reduce the calorie value of dairy products, provide dietary fiber, and promote prebiotic effects, contributing to a healthier diet [8]. This enhancement of the nutritional profile is further supported by the work of González-Tomás et al. (2008), who found that a 7.5% concentration of inulin significantly improves the viscoelastic properties of both whole and skimmed milk samples [9]. From a health perspective, inulin’s role as a prebiotic is particularly noteworthy. Oliveira et al. (2009) reported that inulin influences the acidification kinetic parameters, probiotic counts, pH, and firmness in probiotic fiber-enriched fermented milk [10]. Furthermore, Ahmed and Rashid (2019) highlighted inulin’s function as a fat replacer in dairy products, providing nutritional and therapeutic benefits that contribute to improved health and a reduced risk of lifestyle-related diseases [11]. In product formulation, inulin is appreciated for its texturizing abilities, acting as a bulking agent and gel-former, which is particularly advantageous in reduced-fat food products [12]. Its inclusion in dairy products like Petit Suisse cheese enhances the health appeal and contributes to a creamier and more satisfying mouthfeel [13].

Xanthan gum, a polysaccharide used as a food additive, has a significant impact on the properties of dairy products. Research by El-Sayed et al. (2002) found that xanthan gum increases the viscosity, curd tension, and sensory properties of dairy products, without markedly affecting pH, total solids, or lactic acid bacterial counts [14]. This enhancement of textural qualities is crucial for consumer acceptance and the overall appeal of dairy products. Moreover, xanthan gum is recognized for its stabilizing and thickening properties in various food applications. Its ability to prevent phase separation and maintain probiotic viability in dairy products, as shown by Khodashenas and Jouki (2020), further highlights its functional versatility [15].

Given the potential of buffalo milk and the functional benefits of inulin and xanthan gum, this study aims to develop a novel buffalo Petit Suisse cheese with enhanced symbiotic properties. By incorporating an alternative whey retention method and exploring the synergistic effects of inulin and xanthan gum, the research focuses on a holistic analysis of the cheese’s physicochemical properties, shelf life, viability of lactic acid bacteria, syneresis behavior, and overall quality. This investigation is poised to offer valuable insights into sustainable dairy production while aligning with the growing trend of health-centric, functional foods. The study also seeks to address the gap in understanding the combined effects of these functional ingredients in buffalo milk-based dairy products, potentially setting a new standard in the dairy industry.

## 2. Materials and Methods

### 2.1. Raw Materials

The milk was sourced from Murrah buffalo breed (*Bubalus bubalis*). Milk composition can be viewed in Appendix A. Inulin and xanthan gum were purchased from Sigma-Aldrich (São Paulo, Brazil) from the brand’s website, and the other raw materials were acquired from local markets in Campina Grande, Paraiba, Brazil.

### 2.2. Buffalo Milk Skimming and Fermentation

Fresh milk from the Murrah buffalo breed (*Bubalus bubalis*) was allowed to equilibrate to approximately 10 °C. Using a centrifuge operating at a maximum speed of 4000 rpm, the milk was centrifuged for approximately 20–30 min to ensure optimal cream separation. The duration was adjusted based on visual inspection of separation efficiency. The extracted cream was then collected into sterilized containers and stored at 4 °C until further use. Buffalo skimmed milk was fermented using kefir grains in a sterilized glass container. A proportion of 15 g of kefir grains was used for every 500 mL of milk. The mixture was allowed to ferment for 24 h at room temperature. The kefir grains were separated after fermentation, and the fermented milk was stored at 4 °C.

### 2.3. Production of Buffalo Petit Suisse under a Design of Experiments

The production of the potentially symbiotic buffalo Petit Suisse cheese with whey retention followed this sequence: fat (87.5 g) and sugar (63.7 g) were weighed and added to the buffalo milk (420 g;, the mixture was then homogenized and pasteurized (90 °C for 5 min), and then cooled to 35 °C; 5% fermented kefir milk was added, stirred for two minutes, and then rennet was added according to the manufacturer’s instructions; subsequently, the mixture was incubated at 35 °C in a bacteriological oven for a 24 h period; t the end of this period, the mixture was cooled, processed, and added with the remaining ingredients—xanthan gum, inulin, fruit pulp (126 g), aroma (2.8 g), and potassium sorbate (0.35 g)—one at a time under processing, until fully homogenized; finally, the cheeses were packed in sterile plastic containers and stored at 4 °C. The amounts of xanthan gum and inulin on the final formulation varied according to a full factorial experimental design 2^2^ with three central point experiments (Table 1). The independent variables were inulin (g) and xanthan gum (g), evaluated at three levels (−1, 0 and +1) as indicated in Table 1, totaling 7 experiments.

### 2.4. Physicochemical Analyses

The water content, ash and protein content were determined according to the official methods of AOAC International (2010) [16]. To measure the water content, the cheese samples underwent a drying process in an oven at a 105 °C until constant weight was achieved. The ash content was determined by incinerating the dried samples at 550 °C for 6 h to ensure complete combustion of organic matter, with the remaining residue measured as ash. The protein content assessment involved the Kjeldahl method, which includes digestion of the sample, followed by distillation and titration. The total nitrogen content obtained from this process was then converted to protein content using a 6.30 factor. The total lipid content was determined by the method described by Bligh & Dyer (1959) [17]. Briefly, this method involved a cold extraction process where a mixture of chloroform, methanol, and water was used to extract lipids from the samples. The lipid-containing phase was then separated, dried, and the residual lipid content was weighed.

The determination of total sugars was carried out according to Lane and Eynon (1934) [18]. Briefly, this method required treating the cheese samples with sulfuric acid to hydrolyze polysaccharides into simpler sugars. The reducing sugars released were quantified using a titrimetric method with a known concentration of standard sugar solution for calibration. Total titratable acidity and pH in lactic acid were determined following the methodology of Instituto Adolfo Lutz (2008) [19]. Briefly, the acidity was measured using a titration method where the cheese sample was titrated with a standard sodium hydroxide solution to a specific endpoint pH, and the pH measurement was performed using a calibrated pH meter after homogenizing the samples with distilled water.

### 2.5. Texture Analysis

Texture was analyzed using the AACC method 74-09 (2000) [20], on a texturometer model TA-XT2 (Stable Micro Systems, Surrey, United Kingdom). The probe A/BE—d45 Back Extrusion Rig 45 mm Disc was used, and the set parameters were compression 2 mm/s, distance 25 mm, and contact area 1590 mm^2^ at a temperature of 22 °C. Measurements were performed in triplicate, with the probe being extensively washed first with distilled water. As a result of these experiments, force-time curves were built and analyzed to determine some mechanical parameters (firmness, consistency, viscosity index, and cohesivity) [21].

### 2.6. Shelf-Life Study

During the storage period, samples were stored at 4 °C for 28 days. Analyses were carried out on the product on days 0, 7, 14, 21, and 28, regarding the parameters of pH, acidity, lactic bacteria count, and syneresis using the previously described methods.

#### 2.6.1. Determination of Lactic Bacteria Viability

The procedure for enumerating lactic bacteria were performed according to the following methodology, where 25 g cheese samples were homogenized in 225 mL of peptone saline solution (0.85% NaCl and 0.1% peptone), achieving a 10^−1^ dilution. Serial dilutions were then carried out, and 1 mL of the 10^−3^ and 10^−4^ dilutions were inoculated onto Petri dishes, and then MRS culture medium was added (deep plating). After complete agar solidification, the plates were inverted and incubated at 37 °C for 72 h in a microaerophilic atmosphere. At the end of the incubation period, the number of colony-forming units (CFU/mL) was calculated based on the number of confirmed colonies and the inoculated dilution. A Gram stain was then carried out, followed by the catalase test to confirm the lactic bacteria according to the methodology described by Silva, Junqueira, and Silveira (2001) [22]. Results were analyzed using the following equations. The specific rate of cell death per day (k) was calculated as a first-order reaction (Equation (1)).
(1)$$k=\frac{ln\frac{N_{0}}{N}}{t}$$

In this equation, N refers to the bacterial count at a particular storage period (CFU/mL), N_0_ represents the bacterial count at the beginning of the storage (CFU/mL) and t is the storage time. The specific growth rate (μ) was calculated using Equation (2):(2)$$\mu =\frac{\ln\left(N\right)-ln(N_{O})}{t-t_{0}}$$
where N_0_ and N are viable cell numbers measured within the exponential growth phase at times t_0_ and t, respectively. The percentage of cell survival was defined as shown in Equation (3).
(3)$$Survival\;rate\left(\%\right)=\frac{N}{N_{0}}\times 100$$

Here, N represents the number of viable cells count (CFU/mL) at a particular period and N_0_ denotes the initial viable cell count (CFU/mL) at the beginning.

#### 2.6.2. Syneresis determination

Syneresis was determined according to the methodology proposed by [5], using Equation (4):(4)$$Syneresis\left(\%\right)=\frac{WWh}{WCi}\times 100$$
where WWh (g) is weight of whey separated from the gel after centrifugation and WCi (g) is the initial weight of cheese.

### 2.7. Statistical Analysis

For each obtained response, a multiway ANOVA was carried out through linear regression to verify the influence of the factors on the obtained values and to check if there were significant differences (*p* < 0.05) among the treatments. The regression model used is represented in Equation (5).
(5)y=β0+β1x1+β2x2+β3x1x2

Here, y is the independent variable, βi is the model parameter estimators and xi is the coded factors (independent variables). In cases with a statistically significant difference, response surfaces were generated to visualize the optimization range for process, product, and cost improvements. ANOVA statistical analyses and modelling were performed using Python (version 3.12) [Python Software Foundation, Python Language Reference, available at https://www.python.org]. ANOVA calculations were conducted using the scipy.stats module. The results were evaluated by one-way analysis of variance (ANOVA) and average comparison by Tukey’s test at 5% probability, using GraphPad Prism version 10.0.0 for MacOS GraphPad Software, Boston, MA, USA, www.graphpad.com.

## 3. Results

### 3.1. Proximal Composition

The present work studied the inclusion of inulin and xanthan gum on petite Suisse cheese from buffalo milk, potentially creating a dairy product that is both prebiotic and probiotic. The results of the physicochemical properties are presented in Table 1.

Our experimental analysis revealed a notable correlation between the incorporation of inulin and the water content of cheese. Specifically, the data indicated that higher levels of inulin, as observed in Trials 2 and 4, were associated with an increased water content, resulting in a reduction of total solids. This phenomenon is likely attributable to the hydrophilic characteristics of inulin, which can bind and retain water within the cheese matrix [23]. On the contrary, Trials 1 and 3, which featured lower inulin concentrations, demonstrated decreased water content and correspondingly higher total solids. The influence of xanthan gum on water content was less definitive, with no consistent pattern emerging from the data, thus suggesting that the effects of inulin on water content are more significant than those of xanthan gum in determining cheese composition.

Variations in ash content indicate mineral content was relatively minor across the different trials. The most substantial ash content was recorded in Trial 4, which also contained the highest concentration of xanthan gum. This observation suggests a potential relationship between xanthan gum levels and mineral or ash retention within the cheese. However, the relationship did not exhibit linearity, as variations in xanthan gum content did not consistently correlate with ash content, implying the influence of additional factors on the ash content of the cheese. Similar lack of correlation results were found on white cheese [24].

The study found that the alterations in inulin and xanthan gum levels had minimal impact on the fat and protein content of the cheese, which remained relatively consistent across all trials. This stability indicates that the nutritional profile of the cheese, in terms of fat and protein, is not significantly affected by these additives [25]. Such consistency is beneficial for preserving the nutritional value of the cheese while still allowing for textural modifications through these hydrocolloids.

The content of total sugars exhibited more variability, with the highest sugar content aligning with the trials that had increased amounts of inulin (Trials 2 and 4). Given that inulin is a form of carbohydrate, this result was anticipated. The trials with a higher inulin ratio, specifically 16 g/100 g, showed a substantial rise in sugar content, whereas Trials 1 and 3, with only 6 g/100 g of inulin, presented the lowest sugar levels. This apparent correlation indicates that adjusting the inulin content can effectively modulate the sweetness and possibly the caloric value of the cheese, which can be tailored to meet consumer preferences and nutritional objectives. After modeling, the multiway ANOVA revealed that the total sugar model was significant for *p* < 0.05. Equation (6) presents the first-order model for total sugars.
(6)$$Total\;Sugars\;(q/100\;g)=11.9+0.62\times Inulin+4.78\times Xanthan\;Gum+0.003\;Inulin\times Xanthan\;gum,R^{2}=0.98$$

### 3.2. Mechanical Properties

The results of the texture-dependent mechanical properties are presented in Table 1. The statistical analysis of the firmness data revealed a significant model with a *p*-value of 0.0144. The coefficients indicated a negative influence of inulin (β1 = −0.0003) and a positive influence of xanthan gum (β2 = 0.9067) on firmness, along with a significant interaction term (β3 = 0.1267). The model’s R^2^ value was 0.9580, suggesting a high explanatory power. The positive coefficient for xanthan gum suggests its crucial role in increasing the firmness of the cheese, likely due to its thickening properties. This xanthan gum feature is correlated with its ability for form secondary structures due to this polysaccharide side-chains [26]. The negative coefficient for inulin may indicate its role in softening the cheese texture, which is attributable to its water-binding capacity. Similar behavior was found for caprine fresh cheese with inulin [27]. The interaction term suggests a synergistic effect when inulin and xanthan gum are present. Equation (7) presents the model for firmness.
(7)$$Firmness\left(N\right)=-0.287-0.0003\times Inulin+0.9067\times Xanthan\;Gum+0.1267\;Inulin\times Xanthan\;gum,R^{2}=0.95$$

The consistency model yielded a significant *p*-value of 0.0152. The coefficients showed a strong positive effect of xanthan gum (β2 = 16.7733) on consistency, while inulin had a moderate positive influence (β1 = 0.524). The R^2^ value was 0.9565, indicating the model’s high explanatory power. Xanthan gum’s substantial positive impact on consistency aligns with its role in improving the structural integrity of the cheese. This finding reveals that xanthan gum impacts the stabilization of the whey protein gel network [28]. Inulin’s moderate positive effect might be related to its fiber content, contributing to a denser texture. The interaction term (β3 = 0.41) indicates the combined effect of inulin and xanthan gum on consistency is significant. The model for the viscosity index was highly significant (*p*-value = 0.008934) with an R^2^ value of 0.969559. Xanthan gum had a significant positive effect (β2 = 4.426667), while inulin had a negative impact (β1 = −0.202000). Xanthan gum’s positive influence confirms its role in increasing viscosity, a desirable property for spreadable cheeses. The negative effect of inulin might suggest its role in reducing viscosity, possibly due to its interaction with the water content in the cheese. Similar result was found for Petit Suisse cheese [29]. The high R^2^ value indicates that the model effectively captures the relationship between the ingredients and the viscosity index. Equations (8) and (9) presents the first order model for consistency and viscosity index, respectively.
(8)$$Consistency=-8.051+0.524\times Inulin+16.70\times Xanthan\;Gum+0.41\;Inulin\times Xanthan\;gum,R^{2}=0.95$$
(9)$$Viscosity\;Index=-1.63-0.202\times Inulin+4.42\times Xanthan\;Gum+1.09\;Inulin\times Xanthan\;gum,R^{2}=0.97$$

The cohesivity model was significant (*p*-value = 0.035189) with an R^2^ of 0.923346. Both inulin and xanthan gum showed negative coefficients (β1 = −0.0960 and β2 = −0.7800, respectively), but their interaction had a positive effect (β3 = 0.2300). The negative coefficients for inulin and xanthan gum suggest that higher concentrations of these ingredients independently may reduce cohesivity. However, the positive interaction term indicates they might improve cohesivity when used together, possibly due to a balancing effect between moisture retention (by inulin) and thickening (by xanthan gum). Equation (10) presents the model for cohesivity.
(10)$$Cohesivity=0.4285-0.096\times Inulin-0.78\times Xanthan\;Gum+0.23\;Inulin\times Xanthan\;gum,R^{2}=0.92$$

The interaction between inulin, a soluble fiber, and xanthan gum, a thickening agent, plays a critical role in determining the cheese’s texture and consistency. Inulin’s presence influences the moisture content and interacts with the milk proteins, affecting the protein network and altering the texture, possibly making it softer and affecting water-holding capacity [30]. This interaction is particularly important in terms of the cheese’s syneresis, which impacts consistency over its shelf life. Meanwhile, xanthan gum’s interaction with proteins enhances the cheese’s stability and texture, forming a uniform protein network essential for desired consistency and mouthfeel. Its water-binding properties significantly influence the firmness and spreadability of the cheese [31].

The combined effect of these two ingredients on the cheese’s properties is multifaceted presented by Figure 1. While they individually influence firmness and consistency, their synergistic interaction determines the final texture, ensuring the cheese is neither too firm nor too soft, with optimal spreadability.

### 3.3. Optimization of Petit Suisse Formulation

A statistical approach was employed to optimize the formulation of Petit Suisse, leveraging desirability functions integrated with regression models. The objective was to fine-tune the levels of inulin and xanthan gum to achieve a product profile that aligns with health-conscious consumer preferences and desirable sensory attributes. The optimization focused on achieving lower total sugars (10 g/100 g), a specific range of firmness (3–6 N), optimal viscosity index (13), and higher cohesivity (1.29). The optimization process identified that an inulin level of approximately 7.29 g/100 g and a xanthan gum level of around 1.06 g/100 g would yield the most desirable product attributes. This result was obtained by maximizing the desirability functions corresponding to each product property, namely total sugars, firmness, viscosity, and cohesivity. The overall desirability score at these optimal levels was approximately 0.64 on a scale from 0 (completely undesirable) to 1 (fully desirable). This score reflects a balanced compromise among the various attributes, considering their individual importance and interplay. The desirability functions were designed to prioritize lower sugar content and a texture profile that balances firmness with cohesivity and viscosity, catering to the dual demands of health and sensory appeal.

Figure 2 presents a contour plot that was created to visualize the landscape of desirability across varying levels of inulin and xanthan gum. This plot illustrates how incremental changes in these ingredients’ levels influence the product’s desirability. The optimal point is marked, serving as a guide for formulators. The gradients of desirability across the plot also provide insights into the robustness of the formulation. Regions with gradual color transitions indicate areas where the product quality is less sensitive to ingredient-level variations, which is valuable information for manufacturing and quality control.

### 3.4. Petite Suisse Shelf-Life

#### 3.4.1. pH and Titratable Acidity

Figure 3 presents the pH and TTa time course during the storage period. An increase in total titratable acidity is noticeable from Day 0 to Day 21, followed by a slight decrease or stabilization by Day 28, indicating active fermentation processes in the initial weeks due to lactic acid bacteria. The higher inulin content in Trials 2 and 4 correlated with higher initial acidity and a more pronounced increase over time, suggesting that inulin enhances fermentation activity, likely due to its prebiotic effects [32]. Xanthan gum’s influence was also observed, with different acidity progression in trials with varying gum concentrations. This could be attributed to the xanthan gum’s impact on the cheese matrix, affecting bacterial distribution and activity.

In terms of pH, there were significant changes over the storage period. Initially, the pH values were generally higher, but as the storage time increased, a gradual decrease in pH was observed in most trials, aligning with the increase in acidity levels due to lactic acid production. For example, in Trial 1, the pH decreased from 3.82 on Day 7 to 2.96 on Day 21. Similarly, in Trial 4, the pH decreased from 3.42 on Day 7 to 3.10 on Day 28. These pH changes are indicative of the ongoing microbial activity and the production of lactic acid as the primary fermentation product.

The changes in pH and acidity in the cheese over time are critical for understanding the maturation process and flavor and texture development. The variations across different trials underscore the impact of inulin and xanthan gum concentrations on these processes. Inulin, particularly at higher levels, appears to accelerate the fermentation process, enhancing the growth and activity of lactic acid bacteria [33]. Xanthan gum, while not directly influencing bacterial growth, impacts the cheese environment, potentially affecting the efficiency of the fermentation process.

The consistency of results, shown by the low coefficient of variation percentages, suggests a reliable influence of formulation changes on cheese acidity and pH. This study demonstrates the significant roles of inulin and xanthan gum in fermentation, highlighting their potential in optimizing cheese formulations for desired sensory qualities, textural characteristics, and probiotic benefits. Further research could explore how these changes in acidity and pH impact the cheese’s overall flavor profile, texture, and shelf-life, contributing to the development of high-quality, health-beneficial dairy products [14].

#### 3.4.2. Viable Lactic Acid Bacteria

In the context of the viability of lactic acid bacteria(LAB), also presented in Figure 3, the roles of inulin and xanthan gum are pivotal and interconnected. Additionally, Table 2 presents the results regarding the storage period and the modeling applied to the results to determine the cell death rate, specific growth rate, and cell survival.

The initial counts of viable lactic acid bacteria (CFU/mL) at the outset of the storage were relatively uniform across all trials, ranging from 1.10 × 10^6^ to 1.16 × 10^6^. This consistency provided a standardized baseline for assessing the influence of inulin and xanthan gum on bacterial growth. Over the 28 days, a general trend of increasing bacterial counts was observed in all trials, with Trials 2 and 4 showing higher levels (16 g/100 g), showing a more pronounced increase. By day 28, these trials reached bacterial counts of 1.20 × 10^6^ and 1.23 × 10^6^ CFU/mL, respectively. This trend suggests that increased inulin concentrations may foster a more conducive environment for the growth of lactic acid bacteria, potentially due to its prebiotic properties.

The specific rates of cell death, indicated by negative values across all trials, reflected a decline in the rate of cell death over time. Particularly notable was Trial 4, which combined high levels of inulin (16 g/100 g) and xanthan gum (0.8 g/100 g) and showed the most significant decrease in cell death (−0.00271 day^−1^). This indicates that this specific formulation might be more effective in maintaining cell viability. Correspondingly, the specific growth rates varied across the trials, with Trial 4 exhibiting the highest growth rate (0.00426 day^−1^), aligning with the higher bacterial counts observed. This implies that the combination of higher inulin and xanthan gum concentrations supports an environment favorable for bacterial growth [34].

Furthermore, the cell survival rate, which was over 100% in all trials, indicates the maintenance and proliferation of viable cells over the storage period. Trial 4, with the highest cell survival rate (107.89%), reinforced the notion that higher concentrations of inulin and xanthan gum are advantageous for bacterial growth and survival.

These findings suggest that the formulation of buffalo Petit Suisse cheese can be strategically optimized by adjusting the concentrations of inulin and xanthan gum to enhance the growth and survival of lactic acid bacteria. This has implications for the cheese’s probiotic potential and its overall quality and shelf life. The results provide a valuable foundation for further research into developing functional dairy products that are both nutritious and have an extended shelf life, aligning with the evolving consumer preferences for health-oriented and sustainable food products.

#### 3.4.3. Syneresis

The effect of varying concentrations of inulin and xanthan gum on the syneresis of buffalo Petit Suisse cheese across several trials over a 28-day period is presented in Figure 3. Syneresis, the process of liquid expulsion from cheese, is a crucial factor in determining the cheese’s texture and overall quality. In the first trial with 6% inulin and 0.5% xanthan gum, syneresis was observed to increase significantly from day 0 (14.565%) to peak at day 21 (28.614%), followed by a slight decrease by day 28 (26.669%). This indicates that the cheese matrix was less stable at these concentrations, leading to an increase in liquid expulsion over time. Conversely, the second trial with a higher inulin concentration (16%) but the same level of xanthan gum started with higher initial syneresis (21.917%) on day 0, which then gradually decreased to 17.261% by day 21 before slightly increasing again on day 28 (20.512%). This trend suggests that while higher inulin levels initially contribute to higher syneresis, they might also subsequently stabilize the cheese matrix over time.

Remarkably, Trials 3 and 4, with 6% and 16% inulin combined with a higher xanthan gum concentration (0.8%), exhibited no syneresis over the 28 days. This underscores the effectiveness of xanthan gum at higher concentrations in stabilizing the cheese matrix and preventing the expulsion of liquid.

The control group (Trials 5, 6, and 7) with an intermediate concentration of inulin (11%) and xanthan gum (0.65%) showed no syneresis for the first 14 days. However, a minimal syneresis was recorded thereafter, peaking on day 14 (5.636%) and then gradually decreasing. This pattern indicates that these intermediate concentrations provide a balance, minimizing syneresis while maintaining the quality of the cheese. When retained in the cheese matrix, whey proteins can interact with inulin and xanthan gum, potentially forming a more cohesive and less syneresis-prone structure [35]. This interaction can be attributed to forming a finer protein network, which can better trap and hold water molecules, thus reducing the tendency of the cheese to expel water [36].

The observed patterns of syneresis in the study reflect these macromolecular interactions. In formulations with higher inulin levels but lower xanthan gum (as seen in Trial 2), the initial high syneresis rate that gradually decreases over time could be attributed to the restructuring of the inulin-protein network, influenced by the presence of whey proteins and lactose. This result suggests that weak gels are initially formed and become stronger, reducing the tendency for syneresis [37]. In contrast, the absence of syneresis in trials with higher xanthan gum concentrations suggests that xanthan’s thickening and stabilizing properties are more dominant in these formulations, effectively countering the water expulsion tendency despite whey. This suggestion is aligns well with previous findings that show xanthan gum influence on beta-lactoglobulin adsorption capacity due to biopolymer segregative interactions [38]. The findings suggest that while inulin contributes to syneresis, particularly at higher concentrations, the presence of xanthan gum, especially at higher levels, effectively mitigates this effect, thereby enhancing the textural stability and quality of the cheese. These insights are vital for optimizing cheese formulations to balance texture, quality, and consumer appeal.

## 4. Conclusions

In summary, our research contributes to the field of dairy product development by demonstrating the successful creation of a buffalo Petit Suisse cheese with enhanced symbiotic properties, achieved through an innovative whey retention approach and the incorporation of inulin and xanthan gum. The pivotal findings of this study are the significant roles played by these additives in modifying the cheese’s texture, consistency, and microbial environment. One of the key insights is the positive effect of higher inulin levels on the fermentation process. Our results showed that increased inulin not only enhanced the growth of lactic acid bacteria but also contributed to the overall probiotic quality of the cheese. This suggests a promising avenue for further exploration into the use of inulin in dairy products, particularly for enhancing their probiotic attributes.

Additionally, the role of xanthan gum in reducing syneresis and maintaining the creamy texture of the cheese underscores the importance of texture-modifying agents in dairy product formulation. The interaction between xanthan gum and inulin, along with the whey retention strategy, was found to be crucial in balancing the texture and microbial stability of the cheese. These findings highlight the potential of combining different functional ingredients to optimize the textural and microbiological properties of dairy products. For future research, we recommend a deeper exploration into the synergistic effects of various functional ingredients in dairy formulations. There is also a need to investigate the long-term stability and sensory attributes of such products, which are crucial for consumer acceptance. Furthermore, extending this research to other types of dairy products could provide broader insights into the applicability of these methods in the dairy industry.

## Figures and Tables

**Figure 1 foods-12-04343-f001:**
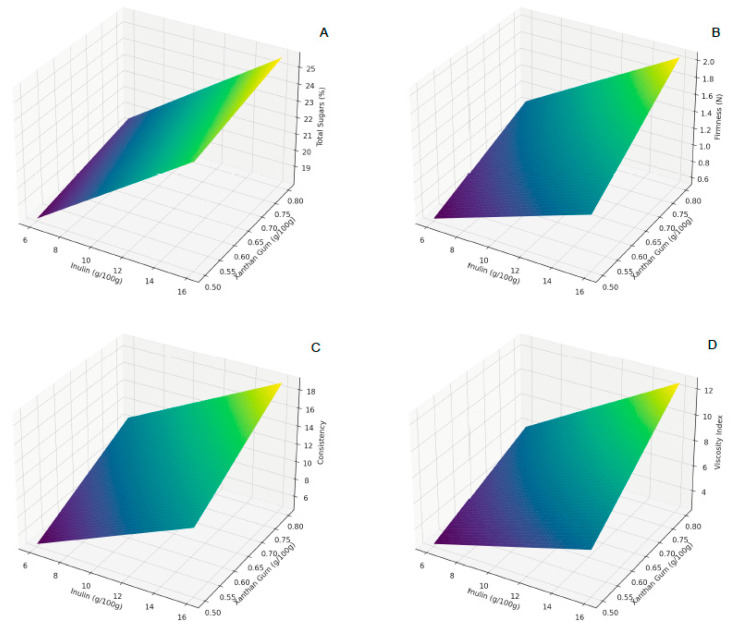
Response surface plots for (**A**) total sugars, (**B**) firmness, (**C**) consistency (**D**) viscosity index.

**Figure 2 foods-12-04343-f002:**
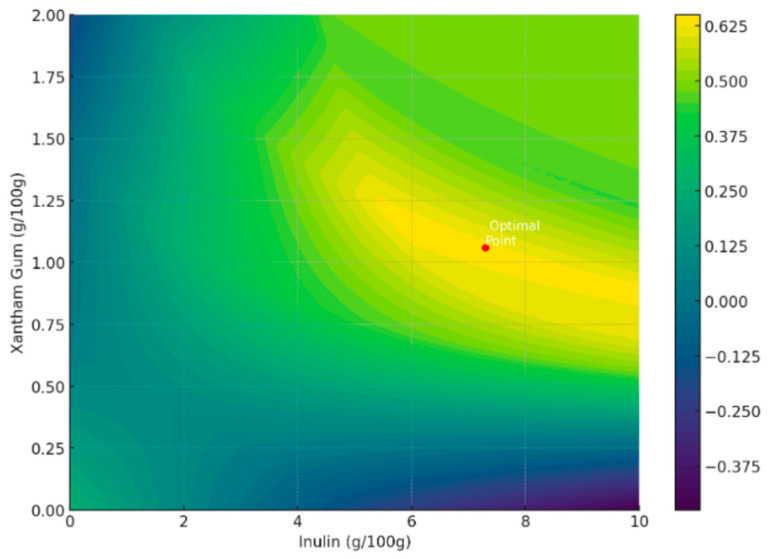
Contour plot evidencing the optimal formulation point.

**Figure 3 foods-12-04343-f003:**
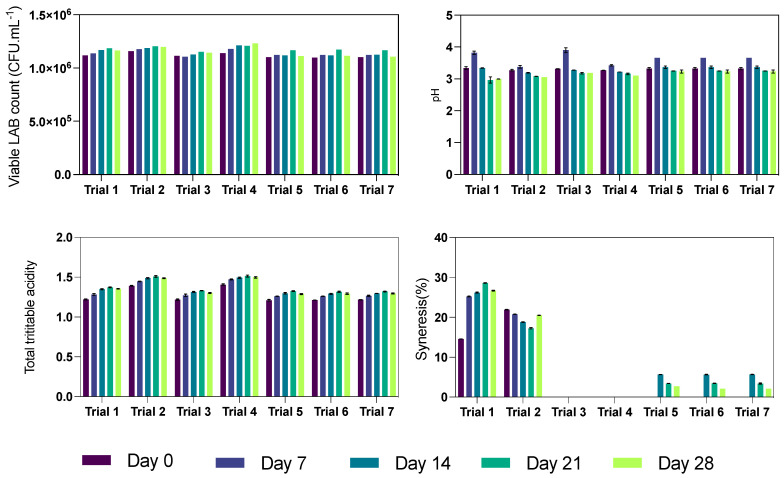
Changes in viable lactic acid bacteria, pH, titratable acidity and syneresis during storage time for each trial.

**Table 1 foods-12-04343-t001:** Formulation, proximal composition and mechanical properties of buffalo Petit Suisse cheese according to the experimental design.

Trials	Units	1	2	3	4	5	6	7
Inulin	(g/100 g)	6 (−1)	16 (+1)	6 (−1)	16 (+1)	11 (0)	11 (0)	11 (0)
Xanthan Gum	(g/100 g)	0.5 (−1)	0.5 (−1)	0.8 (+1)	0.8 (+1)	0.65 (0)	0.65 (0)	0.65 (0)
Strawberry fruit pulp	(g)	126	126	126	126	126	126	126
Strawberry aroma	(g)	2.8	2.8	2.8	2.8	2.8	2.8	2.8
Potassium sorbate	(g)	0.35	0.35	0.35	0.35	0.35	0.35	0.35
Proximal composition								
Water content	(g/100 g)	53.16 ± 0.08	59.23 ± 0.2	51.57 ± 0.3	58.29 ± 0.2	57.70 ± 0.05	57.65 ± 0.08	57.48 ± 0.06
Total Solids	(g/100 g)	46.83 ± 0.08	40.76 ± 0.2	48.42 ± 0.3	43.70 ± 0.2	42.29 ± 0.05	42.34 ± 0.08	42.51 ± 0.06
Ash	(g/100 g)	0.41 ± 0.002	0.45 ± 0.04	0.52 ± 0.05	0.66 ± 0.004	0.46 ± 0.008	0.46 ± 0.01	0.45 ± 0.01
Fat	(g/100 g)	8.33 ± 0.07	8.37 ± 0.07	8.31 ± 0.07	8.31 ± 0.17	8.33 ± 0.11	8.38 ± 0.11	8.37 ± 0.19
Protein	(g/100 g)	3.11 ± 0.01	3.13 ± 0.01	3.13 ± 0.02	3.15 ± 0.04	3.12 ± 0.01	3.12 ± 0.002	3.12 ± 0.007
Total Sugars	(g/100 g)	18.31 ± 0.28	24.5 ± 0.17	19.75 ± 0.22	25.98 ± 0.19	21.50 ± 0.13	21.46 ± 0.13	21.46 ± 0.39
Mechanical properties								
Firmness	(N)	0.61 ± 0.08	1.24 ± 0.03	1.11 ± 0.06	2.12 ± 0.22	1.20 ± 0.02	1.05 ± 0.05	1.10 ± 0.02
Consistency	-	5.39 ± 0.03	12.7 ± 0.34	11.16 ± 0.14	19.68 ± 0.19	10.94 ± 0.04	10.13 ± 0.18	10.85 ± 0.06
Viscosity Index	-	3.01 ± 0.11	6.44 ± 0.16	6.30 ± 0.14	13 ± 1.22	6.74 ± 0.05	5.95 ± 0.06	6.23 ± 0.04
Cohesivity	(N)	0.23 ± 0.005	0.42 ± 0.01	0.41 ± 0.006	1.29 ± 0.36	0.42 ± 0.004	0.39 ± 0.002	0.41 ± 0.005

**Table 2 foods-12-04343-t002:** Evolution of viable lactic acid bacteria during storage and corresponding microbial kinetic parameters.

Trial	Units	1	2	3	4	5	6	7
Inulin	(g/100 g)	6	16	6	16	11	11	11
Xanthan Gum	(g/100 g)	0.5	0.5	0.8	0.8	0.65	0.65	0.65
Viable lactic acid bacteria								
Day 0	CFU/mL	1.12 × 10^6^	1.16 × 10^6^	1.11 × 10^6^	1.14 × 10^6^	1.10 × 10^6^	1.10 × 10^6^	1.10 × 10^6^
Day 7	CFU/mL	1.14 × 10^6^	1.18 × 10^6^	1.11 × 10^6^	1.18 × 10^6^	1.12 × 10^6^	1.12 × 10^6^	1.12 × 10^6^
Day 14	CFU/mL	1.17 × 10^6^	1.19 × 10^6^	1.13 × 10^6^	1.21 × 10^6^	1.12 × 10^6^	1.12 × 10^6^	1.12 × 10^6^
Day 21	CFU/mL	1.18 × 10^6^	1.20 × 10^6^	1.15 × 10^6^	1.21 × 10^6^	1.13 × 10^6^	1.13 × 10^6^	1.13 × 10^6^
Day 28	CFU/mL	1.17 × 10^6^	1.20 × 10^6^	1.14 × 10^6^	1.23 × 10^6^	1.11 × 10^6^	1.11 × 10^6^	1.11 × 10^6^
Microbial kinetic parameters								
Specific Rate of Cell Death (k)	Day^−1^	−0.00156	−0.00121	−0.00095	−0.00271	−0.00032	−0.00032	−0.00032
Specific Growth Rate (µ)	Day^−1^	0.00312	0.00182	0.00128	0.00426	0.00129	0.00129	0.00129
Cell Survival	(%)	104.46	103.45	102.70	107.89	100.91	100.91	100.91

## Data Availability

Data is contained within the article.

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
