# Peer review of "Development and Characterization of Symbiotic Buffalo Petit Suisse Cheese Utilizing Whey Retention and Inulin Incorporation"

_foods, 2023, doi:10.3390/foods12234343_

Round 1

Reviewer 1 Report

Comments and Suggestions for Authors

Petit Suisse is a type of creamy and soft cheese from France that has a delicate and sweet taste with a texture closer to a thick yogurt than cheese. This cheese is produced by coagulating milk using rennet and mesophilic bacteria with the possible addition of other food substances. Petit Suisse is a popular dairy product world-wide that is generally consumed by children but is well received by all age groups.

In fermented dairy products, gel formation is the most important technological property because it is directly linked to the structural product stability. Stabilizers and thickeners are used to improve the gel structure formed in these foods. There is a trend in the food industry to value the use of natural ingredients of plant origin. The textural parameters of fermented dairy products are of marked relevance as they play an important role in the product acceptance by consumers.

In the second position in the ranking of the volume of milk produced worldwide, buffalo milk is relevant both for global feed and agribusiness. The buffalo population in the world is increasing, especially in Asia and Africa.

Inulin is a fructo-oligosaccharide and regarded as a soluble dietary fiber which naturally presents in many plant foods. It can behave as a functional food ingredient also due to its bifidogenic nature, being used more frequently in low fat products. Xanthan gum is a polysaccharide used as a thickener, increasing the low shear viscosity in fluid foods without affecting the viscosity of the food at high shear rates. Xanthan gum is a unique hydrocolloid that exhibits significant yield stress values even at low concentrations, which explains the ability of xanthan solutions to stabilize emulsions.

This study aims to develop a novel buffalo petit Suisse cheese with enhanced symbiotic properties. By incorporating an alternative whey retention method, the research focuses on the cheese physicochemical properties, shelf life, viability of lactic acid bacteria and textural analysis.

Recommendations

L 83 Please include all the ingredients (in grams) for all trials in a table or a technological process figure. The recipe includes fruit pulp. What kind of fruit? Is this not a variable? Please clarify.

L 99 Please present the methods used to analyse and calculate, including the type of equipment used. One of your references is a 1000 page book written in Portuguese.

L 177 Please include in table 1 the three levels (+1, 0 and -1) corresponding to each variable.

L 271 fig. 1, L  299 fig. 2  The plots presented in figures 1 and 2 show inulin contents up to 10%, and you have conducted experiments for 11 and 16%. In the experimental design, the xanthan gum content ranges from 0.5 to 0.8%, and the plots show a corresponding range of up to 2%. Please explain.

Reconsider after major revision

Author Response

L 83 Please include all the ingredients (in grams) for all trials in a table or a technological process figure.

Dear reviewer, table 1 was changed to accommodate your demand.

The recipe includes fruit pulp. What kind of fruit? Is this not a variable? Please clarify.

We used strawberry fruit pulp. Now its mentioned in the manuscript.

L 99 Please present the methods used to analyse and calculate, including the type of equipment used. One of your references is a 1000 page book written in Portuguese.

Dear reviewer, detailed information is now presented.

L 177 Please include in table 1 the three levels (+1, 0 and -1) corresponding to each variable.

Dear reviewer, table 1 was changed to accommodate your demand.

L 271 fig. 1, L  299 fig. 2  The plots presented in figures 1 and 2 show inulin contents up to 10%, and you have conducted experiments for 11 and 16%.

Figure 1 was corrected. We conducted experiments with inulin from 6 to 16%. 

In the experimental design, the xanthan gum content ranges from 0.5 to 0.8%, and the plots show a corresponding range of up to 2%. Please explain.

Figure 1 was corrected. Figure 2 the formulation ranges was slightly increased for a better understanding. I’m sending the version with the ranges we used. As you can see, the optimal point is not clear.

Reviewer 2 Report

Comments and Suggestions for Authors

The manuscript is quite important for science, but requires refinement. 

1 - Introduction – authors must write again. In its current form, it does not encourage interest in the topic, it is too short, it lacks a specific goal and no explanation of why such a topic was undertaken.

2.1. – please indicate the composition of raw milk.

2.5. – please provide the measurement temperature

3 – Table 1, no statistically significant differences?

5 – Needs to be changed. Please focus on the most important assumptions and recommendations for other researchers

Author Response

The manuscript is quite important for science, but requires refinement. 

1 - Introduction – authors must write again. In its current form, it does not encourage interest in the topic, it is too short, it lacks a specific goal and no explanation of why such a topic was undertaken.

Dear reviewer, the introduction was re-written.

2.1. – please indicate the composition of raw milk.

Dear reviewer, you can find the composition of the raw material on the supplementary material, table S1

2.5. – please provide the measurement temperature

Dear reviewer, you can find the temperature on line 110

3 – Table 1, no statistically significant differences?

Dear reviewer, please be more specific regarding your request.

5 – Needs to be changed. Please focus on the most important assumptions and recommendations for other researchers

Dear reviewer we re-written the conclusion

Round 2

Reviewer 1 Report

Comments and Suggestions for Authors

Accept in present form

Reviewer 2 Report

Comments and Suggestions for Authors

Accept in present form